# Wellness in the Schools: A Lunch Intervention Increases Fruit and Vegetable Consumption

**DOI:** 10.3390/nu13093085

**Published:** 2021-09-02

**Authors:** Pamela A. Koch, Randi L. Wolf, Raynika J. Trent, Ian Yi Han Ang, Matthew Dallefeld, Elizabeth Tipton, Heewon L. Gray, Laura Guerra, Jennifer Di Noia

**Affiliations:** 1Department of Health and Behavior Studies, Teachers College, Columbia University, New York, NY 10027, USA; wolf@tc.columbia.edu (R.L.W.); rt2569@cumc.columbia.edu (R.J.T.); mad2258@tc.columbia.edu (M.D.); lag2177@tc.columbia.edu (L.G.); 2Saw Swee Hock School of Public Health, National University of Singapore and National University Health System, Singapore 117549, Singapore; yha2103@columbia.com; 3Department of Statistics, Weinberg College of Arts and Sciences, Northwestern University, Evanston, IL 60208, USA; tipton@northwestern.edu; 4College of Public Health, University of South Florida, Tampa, FL 33612, USA; hlgray@usf.edu; 5Department of Sociology, William Paterson University, Wayne, NJ 07470, USA; dinoiaj@wpunj.edu

**Keywords:** school meals, school-based intervention, fruit and vegetable consumption, elementary school, school wellness, food and nutrition education programming

## Abstract

Wellness in the Schools (WITS) is a national non-profit organization partnering with public schools to provide healthy, scratch cooked, less processed meals (called an Alternative Menu), and active recess. This study examined the effects of WITS programming on school lunch consumption, including fruit and vegetable intake, in second and third grade students in New York City public schools serving a high proportion of students from low-income households. The intervention was evaluated with a quasi-experimental, controlled design with 14 elementary schools (7 that had initiated WITS programming in fall 2015 and were designated as intervention schools, and 7 matched Control schools). School lunch consumption was assessed by anonymous observation using the System of Observational Cafeteria Assessment of Foods Eaten (SOCAFE) tool in the fall of 2015 (Time 0, early intervention) and the spring of 2016 (Time 1) and 2017 (Time 2). There were no baseline data. Data were also collected on the types of entrées served in the months of October, January, and April during the two school years of the study. Across time points, and relative to students in the Control schools, students in WITS schools ate more fruits and vegetables (units = cups): Time 0: Control 0.18 vs. WITS 0.28; Time 1: Control 0.25 vs. WITS 0.31; and Time 2: Control 0.19 vs. WITS 0.27; *p* < 0.001. They also had more fruits and vegetables (cups) on their trays, which included more vegetables from the salad bar. However, students in the WITS schools ate fewer entrées (grain and protein) and drank less milk than students in the Control schools. Compared to the Control schools, WITS schools offered more homestyle entrées and fewer finger foods and sandwich entrees, i.e., less processed food. Students in WITS schools who received the Alternative menu and all of the WITS programming at all data collection time points selected and consumed more fruits and vegetables. Replication studies with randomized designs and true baseline data are needed to confirm these findings and to identify avenues for strengthening the effects of the program on other school lunch components.

## 1. Introduction

The health benefits of a diet rich in fruits and vegetables are well-recognized. A high intake of fruits and vegetables is associated with reduced risks of cardiovascular disease, cancer, and all-cause mortality [1]. A high intake of vegetables is also associated with a reduced risk of weight gain, being overweight, or obesity [2]. Yet, most Americans aged 2 years and above do not consume the recommended amount of fruits and vegetables [3]. In light of the benefits of fruit and vegetable consumption and the fact that childhood dietary patterns are associated with food patterns later in life, encouraging children to eat more fruits and vegetables is a public health priority [4].

School-based interventions are part of the multipronged approach to promote fruit and vegetable intake in children. Most US children attend school for 6 h a day and consume as much as half of their daily calories at school [5]. No other institution provides as much continuous contact with children in the first two decades of life [6,7]. The school setting may therefore provide opportunities for children to be exposed to and learn healthy eating patterns [7].

Wellness in the Schools (WITS) is a national non-profit organization that partners with public schools to provide healthy, scratch cooked, less processed meals (called the Alternative menu), and active recess periods. WITS aims to improve a variety of student outcomes and influence long-term changes in the school culture around healthy living. School lunch and recess are important parts of the school day to help students develop healthy eating and physical activity habits. In New York City (NYC), the location for this evaluation, WITS provided extra resources to schools for school lunch and recess (see the intervention section below). In a prior evaluation of the recess-related outcomes, WITS programming was found to increase physical activity levels during indoor recess and increase ball-like activities in outdoor recess, suggesting that the program has the potential to increase physical activity during school [8]. An evaluation of the impacts of WITS programming on lunch consumption is needed.

There have been improvements to school meals through the increased nutrition standards in the 2010 Healthy Hunger Free Kids Act, such as increased servings of fruits, vegetables, increased amounts of whole grains, and decreased amounts of sodium [9,10]. However, there has not been an increase in scratch cooking or a decrease in pre-prepared processed meals. Little is known about how a change in the menu to include more scratch cooked and less processed foods affects which meal components students have on their trays, and how much of each meal component they eat.

In this study, we examined the effects of the WITS programming on the school lunch consumption of second and third grade students in NYC public schools serving a high proportion of students from low-income households. This study adds to the literature on school-based interventions as it investigated school lunch consumption in schools with two different menus (the Alternative menu versus the standard menu), and also collected data over two school years to measure sustained changes. The aim was to understand how WITS programming impacted students’ school lunch consumption, including their fruit and vegetable intake.

## 2. Materials and Methods

### 2.1. Design

This evaluation used a quasi-experimental, non-randomized, controlled design to examine the impact of the WITS programming over the 2015–16 and 2016–17 school years. The study was approved by the Institutional Review Boards at Teachers College, Columbia University, and the NYC Board of Education.

### 2.2. Intervention

There were two major components of the WITS programming: (1) Cook for Kids and (2) Coach for Kids. Cook for Kids partnered trained culinary school graduates (called WITS Cooks) with cafeteria staff to: (a) transition the school from the standard menu offered by the Office of Food and Nutrition Services (OFNS) to the Alternative menu that WITS created with OFNS (i.e., daily scratch cooked meals and less processed entrées), (b) add or expand the salad bar, (c) eliminate chocolate milk as a beverage option, and (d) add a waterjet station if the school did not already have one. The WITS Cooks were a presence in the school cafeteria and encouraged students to eat the school lunch, as well as to take and eat vegetables from the salad bar. Through WITS Labs and WITS Bits, a series of seasonal cooking and nutrition classes, the WITS Cooks taught children and families how to prepare and cook healthy, tasty, and affordable meals with whole and less processed foods. All the classes in all the WITS schools received four WITS Labs each school year of the intervention. WITS Bits are short 20 min lessons that were offered to the schools. The number of WITS Bits lessons the classes in the WITS schools received was not recorded. The WITS Cook was at the school and interacted with the students every school day during the first school year of the study and was present and interacted with the students three days a week during the second school year of the study in all the schools. Any NYC public school could adopt the Alternative Menu. However, only schools who were working with WITS would have the WITS Cook, the WITS Labs, and WITS Bits.

Coach for Kids used trained fitness professionals to work with students during school recess (both indoors and outdoors) to transform recess from a time of sedentary socializing to a time of increased physical activity and teamwork. As part of Coach for Kids, the students were empowered and supported during recess with WITS anti-bullying initiatives through the teaching of pro-social behaviors.

For the purposes of this study, we focused on if and how WITS programming changed school lunch consumption.

### 2.3. Sample

Seven elementary schools that initiated WITS programming in fall 2015 served as the intervention schools. We recruited seven schools to serve as matched Control schools. That is, for each of the WITS schools there was a Control school that was similar in terms of demographics (% free/reduced priced lunch, % Black students, % Hispanic students, English Language Literacy and Math State Test scores, 2nd grade enrollment, and 3rd grade enrollment) but served the standard menu and did not receive WITS or other programs that changed school meals or school recess. Two of the Control schools became ineligible after the 2015–16 school year. One Control school chose to switch to the Alternative menu and the other school initiated an extensive nutrition education program that included gardening and cooking education. The research team recruited two new Control schools for the 2016–17 school year.

### 2.4. Outcome Measures

School lunch consumption was measured by anonymous observation using a standardized tool called the *System of Observational Cafeteria Assessment of Foods Eaten (SOCAFE)*. The *SOCAFE* tool allowed each observer to record what four students ate for their school lunch. The *SOCAFE* form was adapted from prior observational instruments used to collect data on school meal consumption [11,12], and *SOCAFE* was refined with personal communications with the lead author and her team [13].

*SOCAFE* records the five basic school lunch components (vegetable, fruit, grain, protein, milk) as well as salad bar vegetable items. For each item the student had on his or her lunch tray, the observer marked off what was present. Additionally, if students took items from the salad bar, the items were marked as present on the student’s tray, and the observer recorded the amount in quarter cups, using a visual salad guide that had photographs of quarter cup portions for each of the most common salad bar items. For food items, the amount consumed was recorded using the following 7-point scale: 0%, 10% (bite), 25%, 50%, 75%, 100%, and more than 100%. When comparing the *SOCAFE* scale to the commonly used quarter-waste method (0%, 25%, 50%, 75%, 100%), the 10% and ≥100% increased the sensitivity [14]. Adding the 10% captured the students who at least tried an item and ≥100% captured the students who had more than one portion of an item. For data analysis ≥100% was counted as 1.5 portions.

For milk, the amount consumed was recorded as 0% (milk not opened), some (50%), and all (100%). We did not collect data on the days when white potatoes were served, which were typically served as baked fries. Thus, all vegetables consumed reported in this study were not white potatoes.

Data were collected in the fall of 2015 (Time 0) and the spring of 2016 (Time 1) and 2017 (Time 2). For students in the WITS schools, Time 0 was shortly after the initiation of the Alternative menu (from the first day of the Alternative menu to the 25th day). Thus, for all the WITS schools, Time 0 was the early intervention. That is, early in the transition to the Alternative menu and in receipt of WITS programming. It should be noted that we originally designed the study to collect baseline data prior to the initiation of the Alternative menu and plans were in place for the Alternative menu to begin five weeks into the start of the school year so that baseline data could be collected. However, when we arrived during the first day of school to collected baseline data, to our surprise, the WITS school was already serving the Alternative Menu, and this was the case for all of the WITS schools. In a school district as big as New York City, food was ordered several weeks in advance; thus, the schools needed to continue serving the Alternative menu, despite the limitations that it now posed to this study. This limitation was addressed in part by rigorously matching the Control schools to the school demographics, as described above in Section 2.3.

For each school, we collected data at each second and third grade lunch period for three days at each time point (Time 0, 1, and 2). This allowed for a variety of entrées and vegetables to be served at each school for each data collection time point.

The *SOCAFE* data collection staff participated in three 6 h training sessions, and the third day involved being at a school for three lunch periods to conduct inter-rater reliability (IRR) with another member of the research team. Then the data collection staff conducted IRR with another team member during each data collection timepoint. For the IRR data, the percentage agreement was calculated for items on the tray and for the amount of the food consumed. This was calculated for an adjacent match (± one category on the 7-point consumption scale) and an exact match.

The IRR at Time 0 was 99% for the items on the tray, 98% for the amount consumed (adjacent match), and 90% for the amount consumed (exact match). IRR at Time 1 was 99% for the items on the tray, 97% for the amount consumed (adjacent match), and 83% for the amount consumed (exact match). IRR for Time 2 was 98% for the items on the tray, 95% for the amount consumed (adjacent match), and 91% for the amount consumed (exact match).

A second type of data was collected to describe the types of entrées that were served. We selected three months for this analysis, October, January, and April, and collected data for the two school years of this study (2015–2016 and 2016–2017). These months represented menu variations by seasons to represent fall, winter, and spring, respectively. Each day on the menu was reviewed to classify the entrées as one of three types: finger foods (e.g., mozzarella sticks, pizza, chicken on the bone, empanada), sandwich entrées (note: a sandwich was defined as a grain with a protein between or inside that is picked up to eat, e.g., burritos, tortillas, calzone, Jamaican beef patty, hamburgers), and homestyle (e.g., macaroni and cheese, ravioli, lo mein, pasta fagioli, rice and beans).

### 2.5. Data Analysis

The school lunch consumption data were analyzed using RCT-YES [www.rct-yes.com (29 August 2021)]. RCT-YES estimates the average treatment effects for designs defined by two key design features: “non-clustered” or “clustered”, and “blocked” or “unblocked” [15]. First, this analysis used a “clustered” design as the students assigned to the Control or WITS were clustered at the school level. Second, this study used a “blocked” design as schools were matched in pairs based on demographic variables, as described in Section 2.3. The clustered, blocked design means the schools were fixed. Thus, these results can only be generalized to schools that are similar in demographics. The outcomes from the RCT-YES model reported statistics for the differences between the Control and WITS schools across all three time points on the outcomes of fruit and vegetable consumption, entrée consumption, and milk consumption. We considered *p* < 0.05 as statistically significant.

For the analyses of the types of entrées, we conducted *t*-tests to compare each entrée type between the WITS and Control schools on Microsoft^®^ Excel for Mac Version 16.52.

## 3. Results

### 3.1. Schools

The demographic characteristics of the Control and WITS schools are shown in Table 1. The Standard Mean Difference (SMD) is presented for each characteristic. Both the Control and WITS schools had a high percentage of students eligible for free- or reduced-price lunch, with a higher percentage in the Control schools at 95.1% compared to the WITS schools at 92.3%. The SMD was 0.5 which Cohen [16] considered a medium magnitude of difference. The demographic variables of the percentages of Black and Hispanic students, the ELA mean scale score, and the Math mean scale score all had a SMD close to 0.2, which Cohen [16] defined as a small magnitude of difference. The SMD for second (0.45) and third grade (0.32) enrollment was between a small and medium magnitude of difference, with a larger enrollment in the Control schools than in the WITS schools.

### 3.2. School Lunch Consumption

The intervention outcomes for school lunch consumption are shown in Table 2. When the students in the WITS schools were compared to the students in the Control schools across all three time points, the students in the WITS schools ate more fruits and vegetables than the students in the Control schools (*p* < 0.001; Table 2). This was true across all three time points. The descriptive data revealed that there were overall more fruits and vegetables (cups) on the tray for the WITS vs. the Control students and more students took salad bar vegetable items in the WITS schools than in the Control schools at all three time points. The total amount of fruits, vegetables, and salad (FVS) consumed in cups were Time 0 Control 0.18 vs. WITS 0.28; Time 1 Control 0.25 vs. WITS 0.31; and Time 2 Control 0.19 vs. WITS 0.27. The percentage differential in FVS consumption in the WITS schools was ***higher*** than the Control schools at all time points: 56% at Time 0; 24% at Time 1; and 47% at Time 2. The mean percentage differential was 42% ***more*** fruits and vegetables for the WITS students compared to the Control students. A review of the descriptive data on consumption of fruit, vegetables, and salad bar items showed that most of the fruit and vegetable consumption was fruit for both the WITS and the Control students.

Students in the WITS schools ate fewer entrées (grain and protein) than students in the Control schools (*p* < 0.001; Table 2). This was true at all three time points. The percentages of entrées consumed were Time 0 Control 46% vs. WITS 42%; Time 1 Control 48% vs. WITS 42%; and Time 2 Control 46% vs. WITS 41%. The percentage differential in the entrée consumption in the WITS schools was ***lower*** than the Control schools at all time points: 9% at Time 0, 13% at Time 1, and 11% at Time 2. The mean percentage differential was 11% ***less*** entrées for the WITS students compared to the Control students.

Students in the WITS schools also drank less milk (*p* < 0.001; Table 2). The percentage of the milk consumed was Time 0 Control 38% vs. WITS 29%; Time 1 Control 38% vs. WITS 30%; and Time 2 Control 31% vs. WITS 26%. The percentage differential in the milk consumption of the WITS schools was ***lower*** than Control schools at all time points: 24% at Time 0, 21% at Time 1, and 16% at Time 2. The mean percentage differential was 21% ***less*** milk for the WITS students compared to the Control students.

### 3.3. School Menus

The analysis of the school menus revealed that compared to the Control schools, WITS schools offered significantly more homestyle entrées (Control 20.8% vs. WITS 36.5%), and significantly fewer sandwich entrées (Control 39.4 vs. WITS 30.5). This suggests less processed foods on the Alternative menu offered through WITS programming. There was no difference in finger food entrées (Table 3).

## 4. Discussion

This study demonstrated that WITS programming made significant changes to the consumption of school lunch. Even though this study used a non-randomized design, the schools in the Control group had similar demographics to the schools which were receiving WITS programming. This study was not designed to determine how different parts of WITS programming affected students school lunch consumption. However, it seems likely that the differences found were due to the differences between the standard and the Alternative menu. The differences found for the three outcomes (fruits and vegetables, entrées, and milk) were similar across all three data collection time points. For the WITS schools, Time 0 was during the first weeks of the transition to the Alternative menu and the receipt of WITS programming, and Time 1 and Time 2 were one and two school years after the transition to the Alternative menu.

For fruits and vegetables, students in the WITS schools consumed 41% ***more*** than students in the Control schools. The difference between students in the Control and WITS schools was smallest at Time 1, when students in both the Control and WITS schools had their highest level of fruit and vegetable consumption.

Looking at the data across the time points, we speculate that the initial differences between the Control and WITS schools at Time 0 (early intervention) and maintained at Time 1 and Time 2, were due to the change to the Alternative menu that focused on vegetables and on an enhanced salad bar. However, without baseline data, it is impossible to know if this difference was from the intervention, even with well-matched WITS and Control schools. The magnitude of difference (.1 cups) found in this study is consistent with the magnitude of increase in fruit and vegetable consumption in school-based intervention studies [17], and likely meaningful since school lunch provides 31% of caloric intake for children [18]. Children consistently eat far less than the recommendations for the amount of fruits and vegetables [19]. These findings are encouraging as they suggest a change to the Alternative menu is related to a higher fruit and vegetable consumption.

There are a few cautions related to the interpretation of these findings. First, it was hypothesized that there would be continued improvements in school lunch consumption over the two years with additional WITS programming, which did not occur. Second, the majority of the total amount of fruit and vegetables consumed was comprised of fruit alone. Interventions to increase vegetable consumption have been challenging and many have proposed that this is due to students not liking the taste of vegetables. Two studies in elementary school children demonstrated that presenting vegetables first, before other foods are offered, increased vegetable consumption. However, these interventions were very intensive and short-term, involving one day and three days of a changed menu [20,21]. It may be that intensive programming (e.g., cooking classes and the WITS Cooks talking to students during lunch) is necessary to maintain the initial changes found in this study. Perhaps even more intensive efforts may be necessary for greater increases in fruit and vegetable consumption, especially vegetable consumption.

For the consumption of the entrées, the students in the WITS schools consumed 11% ***less*** than students in the Control schools. This is concerning as it may indicate that the WITS students found the entrées less appealing and were eating less, as school meals have been shown to be healthier than other food students eat [22]. These data indicate that two years of ongoing exposure to these new entrées was not enough to increase consumption. Perhaps the WITS programming that heavily focused on increasing fruits and vegetables needs to be adapted to focus more on all meal components, which has been shown in other studies to be effective at increasing the consumption of novel entrées [23].

For the consumption of milk, the students in WITS schools drank 21% ***less*** milk than students in the Control schools across the time points. As stated above, part of the WITS programming was to eliminate chocolate milk. The reduction we saw is consistent with other research that showed a reduction in milk when chocolate milk was removed [24]. In other studies, that reduction did not compromise the intake of key milk-related nutrients and was shown to achieve a significant decrease in the consumption of added sugars [24].

This study also found that a significant portion of the food on students’ trays was not consumed. When fruits and vegetables were analyzed across all time points and in both the Control and WITS schools, students had an average of 3/4 of a cup of fruits and vegetables on their trays yet were eating on average 1/4 of a cup. This translated to about 66% of what was on their tray not being consumed and becoming food waste. Similarly, about 56% of the entrées and 68% of the milk was not consumed. There have been calls to reduce food waste in schools [25] and guides have been created for students or school administrators to conduct waste audits to determine what was wasted and more importantly, why students were not eating those foods [26]. Such measures could help reduce food waste and lead to students eating more of the nourishing foods served at school lunch.

### 4.1. Limitations and Strengths

This study’s major limitation is that it was a non-randomized design, which was accounted for by matching the Control schools on demographics. However, it is not known if the schools who chose to participate in WITS programming were different in other ways that could have accounted for differences in school lunch consumption, particularly at Time 0. A second limitation is the blocked, clustered design that treated the schools as fixed. Thus, this study focused on particular kinds of school demographics as reported in Table 1. More research is needed to generalize these results to other contexts. A third limitation is was that Time 0 represented early intervention data (after the switch to the Alternative menu) and not true baseline data. A fourth limitation is that we did not use an intention to treat analysis, and instead dropped and replaced the Control schools that switched to the Alternative menu and adopted a major food and nutrition education program. A fifth limitation may be the lack of sensitivity of the *SOCAFE* instrument; even with the addition of the 10% and ≥100% categories, consumption was assessed on a 7-point scale. Despite these limitations, this is the only evaluation of WITS programming on school lunch consumption. The findings therefore add to the limited data on the nutritional outcomes of WITS. Second, this study used an observational form, *SOCAFE*, that had a high IRR. Third, this study collected data one and two years after the intervention to see the impacts over two years, whereas often school meals-based intervention evaluations are short-term.

### 4.2. Research and Policy Implications

School districts across the country are beginning to implement more scratch cooked menus at school meals. Since the time of this data collection, NYC has conducted a pilot of full scratch cooking in two schools [27], and their 10-year Food Policy Plan, Food Forward NYC [28] calls for a movement toward more freshly prepared, culturally appropriate, and scratch cooked meals. Prior research has also shown that the various factors and operational aspects of how lunch periods are run could have an impact on the consumption of lunch meal components [29,30], with simple modifiable aspects such as slicing fruit and a reduction in noise possibly helping to increase fruit consumption [31,32]. Thus, more research is needed to understand how different menus impact what students have on their trays and what they eat. More research is also needed on how to combine evidence-based nutrition education [33] with changes in school meals, in order to increase the nourishing foods that students eat at lunch, and to decrease food waste.

## Figures and Tables

**Table 1 nutrients-13-03085-t001:** School demographics of the Control and the Wellness in the Schools (WITS) Intervention schools.

Demographic Variable	Control Mean **(SD)**	WITS Mean **(SD)**	SMD ^1^
% students qualifying for free/reduced price lunch	95.1% (5.3%)	92.3% (6.3%)	0.50
% Black students	38.4% (31.5%)	40.9% (26.2%)	0.08
% Hispanic students	48.2% (25.2%)	51.3% (29.8%)	0.11
ELA mean scale score ^2^	286.4 (6.7)	288.0 (14.1)	0.15
Math mean scale score ^3^	291.8 (9.4)	294.4 (11.5)	0.26
2nd grade enrollment	106.1 (58.4)	84.9 (38.5)	0.45
3rd grade enrollment	104.0 (57.8)	89.7 (35.4)	0.32

^1^ SMD = Standard Mean Difference, Magnitude of difference Small SMD = 0.2; Medium SMD = 0.5; Large SMD = 0.8 [16]. ^2^ ELA mean scale score = English Language Arts mean Scale Score on New York State test for 3rd grade students in spring 2015 (youngest students who take this test). Range is 147–429, a score of 320 or above is considered proficient at grade level. ^3^ Math mean scale score = Math mean Scale Score on New York State test for 3rd grade students in spring 2015 Range is 137–397, a score of 314 or above is considered proficient at grade level.

**Table 2 nutrients-13-03085-t002:** School meal consumption: Control and Wellness in the Schools (WITS) intervention schools at Time 0, Time 1, and Time 2.

	Time 0 Early Intervention (Fall 2015)			Time 1 One Year Post Intervention (Spring 2016)			Time 2 One Year Post Intervention (Spring 2017)			
	Control %/Mean **(SD)**	WITS %/Mean **(SD)**	Change ^3^	Control %/Mean **(SD)**	WITS %/Mean **(SD)**	Change ^3^	Control %/Mean **(SD)**	WITS %/Mean **(SD)**	Change ^3^	*p*-Value
Fruits on tray (%)	76.4%	73.8%		81.2%	77.8%		75.9%	76.4%		
Vegetable on tray (%)	46.5%	63.5%		40.0%	70.1%		48.0%	61.9%		
Salad on tray (%)	9.6%	15.8%		9.7%	16.1%		7.1%	18.8%		
FVS^1^ on tray (cups)	0.69 (0.37)	0.82 (0.42)		0.70 (0.39)	0.92 (0.48)		0.69 (0.37)	0.83 (0.42)		
Fruits eaten (cups)	0.14 (0.20)	0.20 (0.25)		0.21 (0.25)	0.21 (0.24)		0.16 (0.22)	0.19 (0.24)		
Vegetables eaten (cups)	0.03 (0.11)	0.05 (0.15)		0.03 (0.12)	0.08 (0.19)		0.01 (0.06)	0.05 (0.13)		
Salad bar items eaten (cups)	0.01 (0.06)	0.03 (0.09)		0.01 (0.06)	0.02 (0.09)		0.02 (0.08)	0.04 (0.12)		
FVS eaten ^1,2^ (cups)	0.18 (0.24)	0.28 (0.30)	56%	0.25 (0.27)	0.31 (0.33)	24%	0.19 (0.25)	0.28 (0.28)	47%	<0.001 ^a^
Grains on tray (%)	96.4%	99.2%		90.5%	99.1%		95.6%	98.9%		
Protein on tray (%)	99.6%	99.3%		97.7%	98.8%		98.9%	98.6%		
Grains eaten (portion)	0.48 (0.44)	0.45 (0.44)		0.45 (0.47)	0.42 (0.44)		0.43 (0.47)	0.43 (0.44)		
Protein eaten (portion)	0.44 (0.37)	0.40 (0.36)		0.49 (0.39)	0.42 (0.39)		0.48 (0.38)	0.37 (0.41)		
Entrée eaten ^2^ (portion)	0.46 (0.37)	0.42 (0.37)	−9%	0.48 (0.39)	0.42 (0.36)	−13%	0.46 (0.37)	0.41 (0.41)	−11%	<0.001 ^b^
Total Milk on tray (%)	74.4%	50.6%		72.3%	49.3%		61.1%	47.4%		
Total Milk drank ^2^ (portion)	0.38 (0.39)	0.29 (0.39)	−24%	0.38 (0.38)	0.30 (0.41)	−21%	0.31 (0.36)	0.26 (0.38)	−16%	<0.001 ^c^

^1^ FVS = Fruits + Vegetables + Salad Bar Vegetable Items. ^2^ Average consumption across all students (those who had items on tray and those who did not have items on tray). ^3^ Percentage differential in consumption in Control schools from WITS schools. ^a^ Analysis with RCT-YES found that students in the WITS schools ate more FVS than students in the Control schools across all three time points. ^b^ Analysis with RCT-YES found that students in the WITS schools ate fewer entrées than students in the Control schools across all three time points. ^c^ Analysis with RCT-YES found that students in the WITS schools drank less milk than students in the Control schools across all three time points.

**Table 3 nutrients-13-03085-t003:** Entrée types offered on regular menu in Control schools and Alternative menus in Wellness in the Schools (WITS) intervention Schools.

	Finger Foods Entrées ^§^		Sandwiches Entrées ^£^		Homestyle Entrées ^∞^	
	Control (count)	WITS (count)	Control (count)	WITS (count)	Control (count)	WITS (count)
October 2015	12	14	15	12	9	13
January 2016	10	14	14	9	9	14
April 2016	13	14	13	11	9	13
October 2016	17	10	14	9	3	9
January 2017	17	8	15	10	7	11
April 2017	15	6	12	10	7	13
Total	84	66	83	61	44	73
Percent ^#^	39.8%	33.0%	39.4%	30.5%	20.8%	36.5%
*t*-test	NS		*p* = 0.002		*p* < 0.001	

^§^ Examples of finger foods entrées are mozzarella sticks, pizza, chicken on the bone, and empanadas. ^£^ Examples of sandwich entrées are burritos, tortillas, calzone, a Jamaican beef patty, and hamburgers. ^∞^ Examples of homestyle entrées are macaroni and cheese, ravioli, lo mein, pasta fagioli, and rice and beans. ^#^ Percent is based on the total entrées served for each group (Control = 211 entrées and WITS = 200 entrées).

## Data Availability

The data presented in this study are available on request from the corresponding author. The data are not publicly available due to IRB restrictions.

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
