# Peer review of "Wellness in the Schools: A Lunch Intervention Increases Fruit and Vegetable Consumption"

_nutrients, 2021, doi:10.3390/nu13093085_

Round 1

Reviewer 1 Report

This manuscript describes results of an intervention that involves scratch cooking in elementary school meal programs. The topic is important and the manuscript is generally well done but I found the writing somewhat colloquial in places and the grammar and clarity (technical precision) can be improved. I have a few suggestions below but there are many places where the wording can be tightened up. I believe the manuscript will make a contribution to the literature on school nutrition and children's dietary intake.

Abstract:

Line 18: “among low-income second and third grade students…” Is it known that the students are from households that are lower-income (i.e., eligible for FRPL), or are these “students at schools serving high proportions of students from lower-income households”? subtle wording differences but these are important distinctions—and I do appreciate the focus on higher-poverty schools because there is a huge need to address dietary disparities among socioeconomically-disadvantaged children!

Line 28 “took more salad bar” seemed a bit awkward, can this be rephrased?

Introduction

Line 55 “less processed meals” seems like it needs a hyphen because “less” is a compound modifier (of processed) and without it this could be alternatively interpreted as consuming fewer meals that are processed but I don’t think that’s the meaning here.

Results

Line 170. “First clustered or not clustered” seems to be a sentence fragment?

Discussion

Line 277. I’m not sure it’s typical scientific writing to note that it is disappointing to not see sustained results- this seems more editorializing than is normal for the interpretation of results.

Line 292. “Had drank” is not grammatically correct.

Line 338. “changes in school meals in increase the nourishing…” seems mis-worded?

Author Response

Response to Reviewer 1 Comments

A brief summary

Summary Comments: This manuscript describes results of an intervention that involves scratch cooking in elementary school meal programs. The topic is important and the manuscript is generally well done but I found the writing somewhat colloquial in places and the grammar and clarity (technical precision) can be improved. I have a few suggestions below but there are many places where the wording can be tightened up. I believe the manuscript will make a contribution to the literature on school nutrition and children's dietary intake.

Response to summary: Thank you. The entire manuscript has been thoroughly reviewed and changes have been made to tighten up the wording.

Comments:

Abstract:

Point 1: Line 18: “among low-income second and third grade students…” Is it known that the students are from households that are lower-income (i.e., eligible for FRPL), or are these “students at schools serving high proportions of students from lower-income households”? subtle wording differences but these are important distinctions—and I do appreciate the focus on higher-poverty schools because there is a huge need to address dietary disparities among socioeconomically-disadvantaged children!

Response 1:  Thank you very much for pointing this out. Yes, this is a very important distinction. The second of what you have postulated was the reality. We have edited the text to better communicate that the students were in schools serving a high proportion of students from low income households. Please see the change in line 19 and lines 75 to 76.

Point 2: Line 28 “took more salad bar” seemed a bit awkward, can this be rephrased?

Response 2: We agree that this is awkward. This was changed to, “which included more vegetables from the salad bar.” See lines 29 to 30.

Introduction

Point 3: Line 55 “less processed meals” seems like it needs a hyphen because “less” is a compound modifier (of processed) and without it this could be alternatively interpreted as consuming fewer meals that are processed but I don’t think that’s the meaning here.

Response 3: A hyphen has been added to make this accurate and avoid the alternative interpretation, which as you said is not the meaning here. See line 56.

Results

Point 4: Line 170. “First clustered or not clustered” seems to be a sentence fragment?

Response 4: Thank you. This has been fixed in line 191.

Discussion

Point 5: Line 277. I’m not sure it’s typical scientific writing to note that it is disappointing to not see sustained results- this seems more editorializing than is normal for the interpretation of results.

Response 5: Thank you. This language was changed to state that it was hypothesized that there would be continued improvements with WITS programming, which did not occur. See lines 304 to 306 (sentence that starts with, “First, it was hypothesized”).

Point 6: Line 292. “Had drank” is not grammatically correct.

Response 6: Thank you. The word “had” was removed. See line 325.

Point 7: Line 338. “changes in school meals in increase the nourishing…” seems mis-worded?

Response 7: Thank you. The phrase “order to” was added so it reads “changes in school meals, in order to increase the nourishing…” See lines 371 to 372.

Reviewer 2 Report

A brief summary

In this study, the authors examined the effects of WITS on the school lunch consumption among low-income elementary school students in NYC public schools, and provided the positive effects of the WITS. Many studies have been done, however, to examine the effectiveness of school-based interventions to increase fruits and vegetable consumption among the school-aged children through nutrition education or modifying school lunch. It is recommended to state more clearly the specific merits or distinction of this study from the previous studies (besides the evaluation of WITS).

Comments:

  1. In measuring outcomes (FVS, grains/proteins, milk consumption), this study collected data at three points in time after initiating the intervention, without having data at baseline. As the authors stated in the limitations, Time 0 was not the true baseline data. Without the baseline data, it is difficult to state that the differences in FVS consumption or other outcome measures between the control and WITS schools at Time 0 was due to the Alternative menu and as the intervention effect, as stated in the manuscript (e.g., Abstract L31-3, Discussion L268-71). Please revise the manuscript considering this comment.
  2. Title of the article: This study did not examine the effects of ‘the recess intervention’ of the WITS, thus, it is recommended to delete ‘Recess’ in the title.
  3. L85-97, 2.2 Intervention: I wonder if the intervention schools received almost the same programs provided by the WITS. Please describe more clearly about the programs. Were all of the WITS programs (WITS Cooks, WITS Labs, WITS Bits, and other components) provided to all of the intervention schools? If not, what was the difference in the programs among the intervention schools?
  4. L 138-42, 2.4 Outcome measures: I wonder why the outcome variables were not measured at baseline. If you had the baseline data on FVS consumption, it is clear that significant difference at Time 0 between the control and WITS schools were due to the Alternative menu. Without the baseline data, the difference at time 0 might reflect something else, e.g., the difference in FVS consumption at baseline between the two groups.
  5. L186, L191, 3.1 Schools : ELA meal scale score, footnote 1. SMD = Standard Meal Difference: Please correct spelling errors.
  6. 3.2 School lunch consumption - FVS consumption: Based on Table 2, it was shown that most of the FVS consumption, even in the intervention schools, were from the consumption of fruits. This can be pointed out in the results.
  7. L231-2, 3.2 School lunch consumption - milk consumption: The percentage differential in milk consumption of WITS schools was higher than control schools at all time points: Change to … lower …
  8. L236-8, 3.3 School menus: Table 3 shows the number of menus by entrée types between the WITS and control schools. Please compare the two groups by entrée types and delete the underlined part. WITS schools offered substantially more homestyle entrees than “finger food” or sandwich entrees …
  9. L268-73, Discussion, School lunch consumption regarding FVS: The results (Table 2) showed that FVS consumption was statistically significantly different between the WITS and control schools. I wonder if these differences were practically significant, although the authors state that these differences (about 0.1 cups) was meaningful based on the references 17 & 18. Table 2 showed that the difference between the WITS and control schools was only 0.02~0.05 cups for vegetable consumption, and 0.01~0.02 cups for salad consumption. Besides the statement in L271-3(magnitude of difference is meaningful), please interpret and mention the findings of FVS consumption more specifically, in terms of vegetable and salad consumption, and include the strategies of increasing consumption of vegetables and salads in this study subjects.

Author Response

Response to Reviewer 2 Comments

A brief summary

Summary comments: In this study, the authors examined the effects of WITS on the school lunch consumption among low-income elementary school students in NYC public schools, and provided the positive effects of the WITS. Many studies have been done, however, to examine the effectiveness of school-based interventions to increase fruits and vegetable consumption among the school-aged children through nutrition education or modifying school lunch. It is recommended to state more clearly the specific merits or distinction of this study from the previous studies (besides the evaluation of WITS).

Response to summary: Thank you. We agree this needed to be clearer a sentence was added the last paragraph in the Introduction which explains the purpose of the study. See lines 75 to 78, which states, “In this study, we examined the effects of the WITS programming on the school lunch consumption of second and third grade students in NYC public schools serving a high proportion of students from low-income households. This study adds to the literature on school-based interventions as it investigates school lunch consumption with schools on two different menus (Alternative menu versus standard menu), and also is collects data over two school years to measure sustained changed.

Comments:

Point 1: In measuring outcomes (FVS, grains/proteins, milk consumption), this study collected data at three points in time after initiating the intervention, without having data at baseline. As the authors stated in the limitations, Time 0 was not the true baseline data. Without the baseline data, it is difficult to state that the differences in FVS consumption or other outcome measures between the control and WITS schools at Time 0 was due to the Alternative menu and as the intervention effect, as stated in the manuscript (e.g., Abstract L31-3, Discussion L268-71). Please revise the manuscript considering this comment.

Response 1: Yes, we agree that this needs to be made clearer. In the abstract see line 24, which states, “There was no baseline data.” Also see lines 33 to 35, which states, “Students in WITS schools, who were receiving the Alternative menu and all of WITS programming at all data collection time point selected and consumed more fruits and vegetables. Replication studies with randomized designs and true baseline data are needed to confirm these findings and to identify avenues for strengthening program effects on other school lunch components.”

 in the Discussion, see lines 295 to 296, which states, “However, without baseline data, it is impossible to know if this difference was from the intervention, even with well-matched WITS and Control schools.”

Point 2: Title of the article: This study did not examine the effects of ‘the recess intervention’ of the WITS, thus, it is recommended to delete ‘Recess’ in the title.

Response 2: The Title has been changed as suggested. The reason that we had “recess” in the title was because our design did not allow us to separate of the lunch and recess programming since all of the WITS schools received both. While we speculated the changes in lunch were due to the lunch components of WITS programming, it was impossible for us to know for sure. Therefore, we agree that removing “recess” for the title is clearer for the data reported in this article.

Point 3: L85-97, 2.2 Intervention: I wonder if the intervention schools received almost the same programs provided by the WITS. Please describe more clearly about the programs. Were all of the WITS programs (WITS Cooks, WITS Labs, WITS Bits, and other components) provided to all of the intervention schools? If not, what was the difference in the programs among the intervention schools?

Response 3: Thank you for asking for these additional details. All of the WITS schools had the WITS Cook present and interacting with students five days a week during the first school year and three days a week during the second school year. All classes received four WITS Labs during both school years. The number of WITS Bits, which are 20-minute lessons, that each class received was not recorded. This is noted in lines 100 to 105.

Point 4: L 138-42, 2.4 Outcome measures: I wonder why the outcome variables were not measured at baseline. If you had the baseline data on FVS consumption, it is clear that significant difference at Time 0 between the control and WITS schools were due to the Alternative menu. Without the baseline data, the difference at time 0 might reflect something else, e.g., the difference in FVS consumption at baseline between the two groups.

Response 4: We agree with the reviewer that the study design would have been much stronger with baseline data and had originally designed the study to collect baseline data prior to the initiation of the Alternative menu. Numerous conversations with the Department of Education Office of Food and Nutrition Services occurred in order to ensure the alternative menu would not begin until five weeks into the start of the school year. However, when we arrived during the first day of school to collect baseline data, to our surprise, the WITS school was already serving the Alternative Menu, and that this was the case for all of the WITS schools. In a school district as big as New York City, food was ordered several weeks in advance, thus the schools needed to continue serving the Alternative menu, despite the limitations that it now posed to our study. We hope that this limitation is in part addressed by rigorously matching the Control schools on % students qualifying for free/reduced price lunch, % Black students, % Hispanic students, ELA man scale score (state tests), Math mean scale score (state tests), 2nd grade enrollment, 3rd grade enrollment. Regardless we realize we do not know for certain that the students’ school lunch consumption changes were due to the WITS Programming versus some other unmeasured factor. We have added this to the manuscript. See lines 154 to 163.

Point 5: L186, L191, 3.1 Schools : ELA meal scale score, footnote 1. SMD = Standard Meal Difference: Please correct spelling errors.

Response 5: These spelling errors are fixed. Lines 209 and 214.

Point 6: 3.2 School lunch consumption - FVS consumption: Based on Table 2, it was shown that most of the FVS consumption, even in the intervention schools, were from the consumption of fruits. This can be pointed out in the results.

Response 6: This is added, please see lines 233 to 235.

Point 7: L231-2, 3.2 School lunch consumption - milk consumption: The percentage differential in milk consumption of WITS schools was higher than control schools at all time points: Change to … lower …

Response 7: This is changed, please see line 258.

Point 8: L236-8, 3.3 School menus: Table 3 shows the number of menus by entrée types between the WITS and control schools. Please compare the two groups by entrée types and delete the underlined part. WITS schools offered substantially more homestyle entrees than “finger food” or sandwich entrees …

Response 8: We have calculated T-tests to compare each entrée type for Control and WITS schools. This is added to data analysis. See lines 200 to 201, and in the results section. See lines 262 to 266.

The Alternative menu (WITS had significantly fewer sandwiches and more homestyle entrées than the Control menu. There was no difference in finger food entrées.

Point 9: L268-73, Discussion, School lunch consumption regarding FVS: The results (Table 2) showed that FVS consumption was statistically significantly different between the WITS and control schools. I wonder if these differences were practically significant, although the authors state that these differences (about 0.1 cups) was meaningful based on the references 17 & 18. Table 2 showed that the difference between the WITS and control schools was only 0.02~0.05 cups for vegetable consumption, and 0.01~0.02 cups for salad consumption. Besides the statement in L271-3(magnitude of difference is meaningful), please interpret and mention the findings of FVS consumption more specifically, in terms of vegetable and salad consumption, and include the strategies of increasing consumption of vegetables and salads in this study subjects.

Response 9: Thank you. We agree and have added more on this in the discussion and also added two more references that focused specifically on increasing vegetable consumption in school meals by offering vegetables alone first. See lines 304 to 312, which states, “There are a few cautions to interpreting these findings. First, it was hypothesized that there would be continued improvements in school lunch consumption over the two years with additional WITS programming, which did not occur. Second, majority of the total amount of fruit and vegetables consumed were comprised of fruit alone. Interventions to increase vegetable consumption have been challenging and many have proposed this is due to students not liking the taste of vegetables. Two studies in elementary school children demonstrated that presenting vegetables first, before other foods are offered, increased vegetable consumption. However, these interventions were very intensive and short-term, one day and three days of a changed menu [20,21].”
